# A character-based analysis of impacts of dialects on end-to-end Norwegian ASR

**Phoebe Parsons**[1]        **Knut Kvale**[2]        **Torbjørn Svendsen**[1]        **Giampiero Salvi**[1,3]

[1]Department of Electronic Systems, NTNU, Trondheim, Norway
[2]Telenor Research, Oslo, Norway
[3]KTH Royal Institute of Technology, EECS, Stockholm, Sweden
{phoebe.parsons, torbjorn.svendsen, giampiero.salvi}@ntnu.no, knut.kvale@telenor.com

## Abstract

We present a method for analyzing character errors for use with character-based, end-to-end ASR systems, as used herein for investigating dialectal speech. As end-to-end systems are able to produce novel spellings, there exists a possibility that the spelling variants produced by these systems can capture phonological information beyond the intended target word. We therefore first introduce a way of guaranteeing that similar words and characters are paired during alignment, thus ensuring that any resulting analysis of character errors is founded on sound substitutions. Then, from such a careful character alignment, we find trends in system-generated spellings that align with known phonological features of Norwegian dialects, in particular, "r" and "l" confusability and voiceless stop lenition. Through this analysis, we demonstrate that cues from acoustic dialectal features can influence the output of an end-to-end ASR systems.

## 1  Introduction

Automatic Speech Recognition (ASR) has, like all machine learning tasks, struggled with generalization. That is, a model will perform well on the task and data it was trained on but when presented with new examples, especially examples that differ in some dimension from the training data, the model will perform markedly less well. In the task of ASR, this means that models often struggle with generating correct transcriptions for speakers whose age, gender, or dialect differs from that of the speakers on which the model was originally trained. Of specific focus in this paper is the impact of dialect on a modern ASR system.

Dialect information has been used in different ways in ASR. In some applications, such as Dialect Identification (DID), the goal is to correctly identify the dialect for a given sample of speech. Hämäläinen et al. (2021), for example, used a combination of speech and text features to perform DID. In other cases, DID is combined with ASR systems to improve transcription accuracy. For example, Zhang and Hansen (2018) used bottleneck features extracted via unsupervised deep learning to perform DID for both Chinese and Arabic. Similarly, Imaizumi et al. (2022) used a multitask model for both DID and ASR. This multitask approach outperformed the single task systems on both DID and ASR.

Beyond DID, the behavior of ASR systems has been analyzed with respect to dialectal speech (as we do in this paper). This in order to explore phonetic phenomena, as well as to gain insights into the way those complex systems work. In these studies, even when dialectal information is not an explicit target, there is still an interest to understand what phonetic and dialectal information has been captured in ASR models. With traditional ASR models, this investigation has been fairly straightforward as these models have consisted of three semi-independent components: the acoustic model, the language model, and the lexicon. Because of the separate acoustic models within these multi-component models, one could, for example, perform clustering on the model parameters themselves such as (Salvi, 2003a,b, 2005). In this work, Salvi performed clustering on the acoustic model features and correlated the resulting clusters with known dialectally realized phonemes. Instead of directly using an acoustic model from an ASR system, Chen et al. (2014) adapted the concept of an HMM acoustic model to automatically discover dialect-specific phonetic rules.

Unlike multi-component ASR systems, investigating modern, end-to-end models for phonetic and dialectal information is quite different.

Whereas parameters from an acoustic model may be extracted and used independently, the acoustic information in an end-to-end model cannot be so easily excised. This design makes it more challenging, but not impossible, to investigate what acoustic information is captured where in the network. Belinkov and collaborators used the output from each layer of an end-to-end system to train phonetic, grapheme, and articulatory classifiers (Belinkov and Glass, 2017; Belinkov et al., 2019). Prasad and Jyothi (2020) investigated dialectal information captured by an end-to-end system using not only layer-wise classification but also gradient and information-theoric analysis. All of these works are focused on analyzing the network-internal representations detached from actual network output.

The output from ASR models is constrained by the model architecture. Traditional ASR models with lexicons are bound to output only words contained within that lexicon. This means that all transcripts generated by these models contained only real, known words even if the transcribed output did not necessarily match the word that was spoken. Additionally, these models do not allow for acceptable variation in spelling. For example, the word, "favorite," would always be spelled "favorite" never "favourite," even if the latter might better reflect the preference of a British English speaker. Conversely, these newer end-to-end architectures, trained using connectionist temporal classification (CTC) loss, produce output at the character instead of word level. This permits the model to create novel words and spellings, potentially better reflecting the phonetic realization of the spoken word. Given that CTC models are allowed to generate novel spellings, there exists the potential that dialectal information will be captured by the model output itself via non-standard spellings.

The goal of this paper is to investigate whether dialectal acoustic information can impact spellings with an end-to-end model. In order to test this, we used wav2vec 2.0 (Baevski et al., 2020) to generate transcriptions of Norwegian speech. We then performed an analysis of the resulting transcripts for captured dialectal knowledge via a dialectal-region based evaluation of character error patterns. From this analysis we are able to see known Norwegian dialectally-based phonological patterns, specifically around "r" and "l" confus-

ability and stop consonant voicing. Thus we illustrate that strong enough acoustic dialectal cues can effect the character output of an end-to-end ASR system.

## 2 Norwegian language and dialects

In this paper, we focus our analysis on the Norwegian language. Though spoken by a relatively small population of a little over 5 million speakers, Norwegian contains many dialects differentiated in phonology, syntax, and lexicon. In addition to dialectal variation, Norwegian also maintains two official written standards: Bokmål and Nynorsk; though neither written standard directly corresponds with a spoken variant. Furthermore, Norway does not recognize any official language standard. Indeed, people are encouraged to use their preferred written standard and native dialect in all aspects of work and life.

The variety in dialects stems from Norway's challenging and rugged topography that has historically forced the populace to organize into many, smaller communities. Over time, the diversity we see in Norwegian dialects developed in these small, isolated communities. As described by phoneticians, there now exist large dialectal phonetic variations ranging from infinitive verb endings to palatalization of consonants, to /r/ and /l/ realizations, to the various pronunciations for the personal pronoun for "I", *jeg* —ranging from [jæi] to [eg] to [i] and more (Skjekkeland, 1997).

While the number of specific Norwegian dialects is quite large, we can group these dialects into larger dialect groups for the purpose of this investigation. These grouping could be either into the regional names used by Skjekkeland or into the even larger, cardinal regions of "East," "West," "North," "South," and "Mid." The analysis outlined in this paper relies on these cardinal regions.

## 3 Methods

### 3.1 Experimental setup and data

In order to investigate the impact of dialect on an end-to-end ASR system, a well-performing baseline model was required. Therefore, we used three models trained by the Norwegian National Library AI Lab and released publicly on the Hugging Face

repository for our analysis [1][2][3]. The first model contained one billion parameters and was originally trained on the XLS-R (Babu et al., 2021). It was then fine tuned using the Norwegian Parliamentary Speech Corpus (NPSC) to transcribe Norwegian Bokmål text. The other two models were fine tuned from the 300 million parameter VoxRex model (Malmsten et al., 2022). One of these 300 million parameter models was fine-tuned to transcribe Bokmål, the other Nynorsk. All models use a 5-gram word-based language model. In all cases, the NPSC corpus was used to fine-tune the models (Solberg and Ortiz, 2022). When evaluated against the NPSC corpus, the Norwegian AI lab reports a word error rate (WER) of 6.33% for the 1 billion parameter model, 7.03% for the 300 million parameter Bokmål model, and 12.22% for the Nynorsk model. These results indicate that these models will make excellent candidates for our analysis.

As stated earlier, the models to be used were trained on the NPSC. This consists of recordings from the Norwegian Parliament and thus the speech style can be considered mostly spontaneous, with perhaps slightly more planning than everyday speech. For analysis purposes, the NPSC was excluded. This is due to data sparsity in the NPSC test set. While the whole test set is acceptable for model evaluation, data becomes untenably sparse when considered dialect-by-dialect. Thus our analysis focuses on results from two unrelated and more dialectally robust corpora: Rundkast and NB Tale.

The Rundkast corpus consists of radio broadcasts from the Norwegian Broadcasting Corporation (NRK) (Amdal et al., 2008). These transcripts are in both Bokmål and Nynorsk which are treated separately for analysis in this paper. Dialectal annotations were added by the transcribers during corpus creation and are provided directly in the speaker metadata.

NB Tale is publicly available from the National Library of Norway's Language Bank and consist of recordings and transcripts of native and non-native speakers of Norwegian. All speech was transcribed using the Bokmål standard. Read speech was recorded from both the native and non-native speakers whereas spontaneous speech was only recorded for the native speakers. For the analysis in this paper only speech from the native speakers was used. For each speaker biographical information was collected, including the municipality in Norway where they lived as a child. From this municipality, a manual mapping to dialect was devised. This mapping then allowed us to infer the speaker's most likely dialect.

Data was prepared and standardized according to the scripts provided in the combined data set, as described by (Solberg et al., 2023). This converted all audio to a mono, 16kHz format. The text was normalized such that capitalizations, punctuation, and hesitations were removed. Additionally, all non-standard forms were converted into a standard equivalent.

## 3.2 Word and character alignment

As our investigation into dialectal impact revolves around analyzing trends in character errors, we require an alignment between reference text and model-generated hypothesis text where words that only differ by a few characters are prioritized for alignment. While character error rate (CER) computed across a whole utterance is useful in understanding an aggregate of character errors, this method loses awareness of word boundaries. For example, "også kalt" and "og såkalt" would be aligned in whole-utterance CER with an insertion and a deletion of a space (resulting in "og så kalt"). However, we prefer an alignment where we recognize that "så" was removed from the first word and "så" as added to the second word. Thus CER, as it is generally used across entire utterances, does not answer for our analysis purposes.

With traditional, word-level Levenshtein-based alignments, word similarity is not considered. Any pair of words that do not exactly match are treated as completely different. However, by considering word similarity, the resulting alignments can be used for analysis of broad trends of spellings (e.g., a word ending in "a" instead of "e") that can indicate dialectal impact.

To accomplish such an alignment, an extension to the traditional Levenshtein alignment was developed (Levenshtein, 1965). Typically edit costs are fixed at a value before alignment is computed. However, in our solution instead of a fixed cost for substitutions, we allow it to be dynamically

---

[1] https://huggingface.co/NbAiLab/nb-wav2vec2-1b-bokmaal
[2] https://huggingface.co/NbAiLab/nb-wav2vec2-300m-bokmaal
[3] https://huggingface.co/NbAiLab/nb-wav2vec2-300m-nynorsk

computed as the CER between the two candidate words. This still ensures that there is no cost for aligning words that are the same while also preferring substitutions of similarly spelled words.

|       | voiced | class | nasal | place | rounding |
|-------|--------|-------|-------|-------|----------|
| "k"   | 0      | 0     | 0     | 5     | 0        |
| "g"   | 1      | 0     | 0     | 5     | 0        |
| "n"   | 1      | 0     | 1     | 2     | 0        |

Table 1: Example of the vectors for "k", "g", and "n" for Norwegian. Indexes of the vector represent features and values represent their realization.

|       | height | front | rounding |
|-------|--------|-------|----------|
| "a"   | 2      | 0     | 0        |
| "e"   | 1      | 2     | 0        |

Table 2: Example of the vectors for "a", and "e" for Norwegian. Indexes of the vector represent features and values represent their realization.

Once word-level alignment is computed using the dynamic substitution cost, we can investigate spelling errors. To ensure characters within a word are aligned optimally, we continue to use the dynamic substitution cost idea and compute the substitution cost between characters as the Euclidean distance between two feature vectors. To support this, articulatory feature vectors were created for each letter in the Norwegian alphabet using the International Phonetic Alphabet (IPA) charts as a guide. Articulatory features were considered as indexes in the vector and the values correspond to the realization. For our work, consonants (see examples in Table 1) were defined and treated separately from vowels (see examples in Table 2). As the goal with these vectors is not to create an accurate grapheme-to-phoneme mapping, nor to perfectly illustrate all possible IPA nuance, but instead to align letters in a more logical way, these vectors were sufficient.

To illustrate the necessity of these vectors, consider the word pair of *inngang* (meaning "entrance") and *enkel* ("easy"). Using a traditional alignment method [4], where all characters substitutions have the same cost, an alignment like in

| reference  | i | n | n | g | a | n | g |
|------------|---|---|---|---|---|---|---|
| hypothesis | e | n |   |   | k | e | l |

Table 3: A possible alignment between *inngang* and *enkel*, generated without accounting for character similarity.

| reference  | i | n | n | g | a | n | g |
|------------|---|---|---|---|---|---|---|
| hypothesis | e | n |   | k | e | l |   |

Table 4: A possible alignment between *inngang* and *enkel*, generated by accounting for character similarity.

Table 3 is generated. However, using articulatory features as a distance, we are able to generate the alignment in Table 4 where "g" and "k" (only differing by voicing), "a" and "e" (both being front vowels), and "n" and "l" (both being sonorants) are aligned.

While this solution is slightly phonologically flawed —wholly ignoring the di- and trigraphs that exist in Norwegian and instead treating the component letters individually, for example —these feature vectors do accomplish the goal of creating a logical character-level alignment. With confidence in our word and character alignment we can perform the investigation into character substitution trends that constitutes our results.

## 4 Results

### 4.1 WER by dialect

To first understand the general trend in recognition across dialects, the WER was calculated for each dialect across the whole of the Rundkast and NB Tale corpora. Transcriptions were generated using both the 300 million and 1 billion parameter Bokmål models for both corpora. Rundkast was further transcribed with the 300 million parameter Nynorsk model (since Rundkast actually contains Nynorsk utterances, unlike NB Tale).

As displayed in Table 5 that shows WER across both corpora and dialects, we can see WER values ranging from the low teens to nearly 40%. These values are markedly higher than the 6.33% WER that was reported on the NPSC which highlights the impact of domain mismatch on ASR; models trained on one domain (the Norwegian Parliament) do not generalize well to new domains (radio and studio recordings).

---

[4]Alignment generated using the Python Levenshtein package: https://github.com/maxbachmann/python-Levenshtein

| Dataset | Dialect | Utterances | Speakers | WER% | | |
|---|---|---|---|---|---|---|
| | | | | 1B Bok | 300M Bok | 300M Ny |
| NB Tale —Bokmål utterances | Other | 5087 | 120 | 25.79 | 26.04 | — |
| | West | 4064 | 93 | 20.34 | 20.78 | — |
| | Mid | 1789 | 40 | 18.14 | 20.02 | — |
| | North | 2760 | 68 | 17.89 | 18.54 | — |
| | South | 591 | 14 | 16.86 | 18.00 | — |
| | East | 1898 | 42 | 16.44 | 17.15 | — |
| Rundkast —Bokmål utterances | Unknown | 199 | 12 | 19.54 | 18.28 | 38.82 |
| | West | 7526 | 176 | 18.21 | 16.66 | 36.28 |
| | Mid | 2917 | 124 | 17.06 | 17.35 | 37.30 |
| | North | 2941 | 153 | 16.38 | 16.13 | 35.31 |
| | South | 1372 | 56 | 16.16 | 15.11 | 35.67 |
| | East | 51303 | 993 | 13.93 | 13.35 | 36.04 |
| Rundkast —Nynorsk utterances | South | 355 | 15 | 31.63 | 30.46 | 31.89 |
| | Mid | 77 | 1 | 30.41 | 29.46 | 27.89 |
| | West | 6024 | 161 | 29.35 | 28.26 | 23.99 |
| | East | 2802 | 34 | 28.27 | 26.96 | 20.49 |
| | North | 13 | 1 | 26.43 | 27.86 | 18.12 |
| | Unknown | 3 | 3 | 0.00 | 0.00 | 0.00 |

Table 5: WER for Rundkast and NB Tale corpora. Transcribed using the all models. As there is no Nynorsk text in the NB Tale corpus, we did not evaluate the Nynorsk model. The WER reported for the models on the NPSC corpus are 6.33% for the 1B model, 7.03% for the 300M Bokmål model, and 12.22% for the 300M Nynorsk model.

For the Bokmål text in both corpora, we can see that models perform best on the "East" dialect region whereas the "West" region has the worst performance. It is unclear which model is generally the best. The 1 billion parameter model performs better than the 300 million parameter model on the NB Tale text, but the 300 million parameter model outperforms the 1 billion on the Rundkast text.

With the Rundkast corpus, we can see that the Bokmål models perform, as expected, poorly on the Nynorsk text with the converse (Nynorsk model evaluated against Bokmål text) being true as well. However, even when the Nynorsk model is evaluated against Nynorsk text, the results are still worse than the Bokmål model of the same size evaluated against Bokmål text.

Of more concern than model accuracy, however, is data scarcity for Nynorsk text. Given that Nynorsk is primarily used in the western part of Norway, the nearly equal split of speakers between Bokmål and Nynorsk for the "West" region is understandable. Moreover, for the other regions ("North" and "Mid" in particular) there are too few speakers to draw conclusions from. Therefore, as we move forward with the character-based analy-

sis, we will be focusing on the Bokmål models and their performance on the Bokmål text.

## 4.2 /r/ and /l/ confusiblity

In Norwegian, /r/ is generally realized as either a voiced apical tap or a voiced velar approximant (Kvale and Foldvik, 1992). These two different pronunciations are considered dialect features, with the approximant version predominating in the "South" and "West" of the country and the tap being the norm in the rest of country. The maps in (Kvale and Foldvik, 1999) and (Skjekkeland, 1997) nicely illustrate this distribution.

Similar to the Norwegian /r/, which can be realized in several variants, the Norwegian /l/ also has dialectally motivated realizations. Many speakers in the "East", "Mid", and southern part of the "North" region of the country produce a voiced retroflex flap. The norm for speakers in the rest of the country ("West", "South", and the remaining part of the "North") is a voiced dental/alveolar lateral (Kvale and Foldvik, 1995).

Understanding these phonetic realizations, we can anticipate that the tapped [ɾ] and the lateral approximant [l] should be minimally confusing for

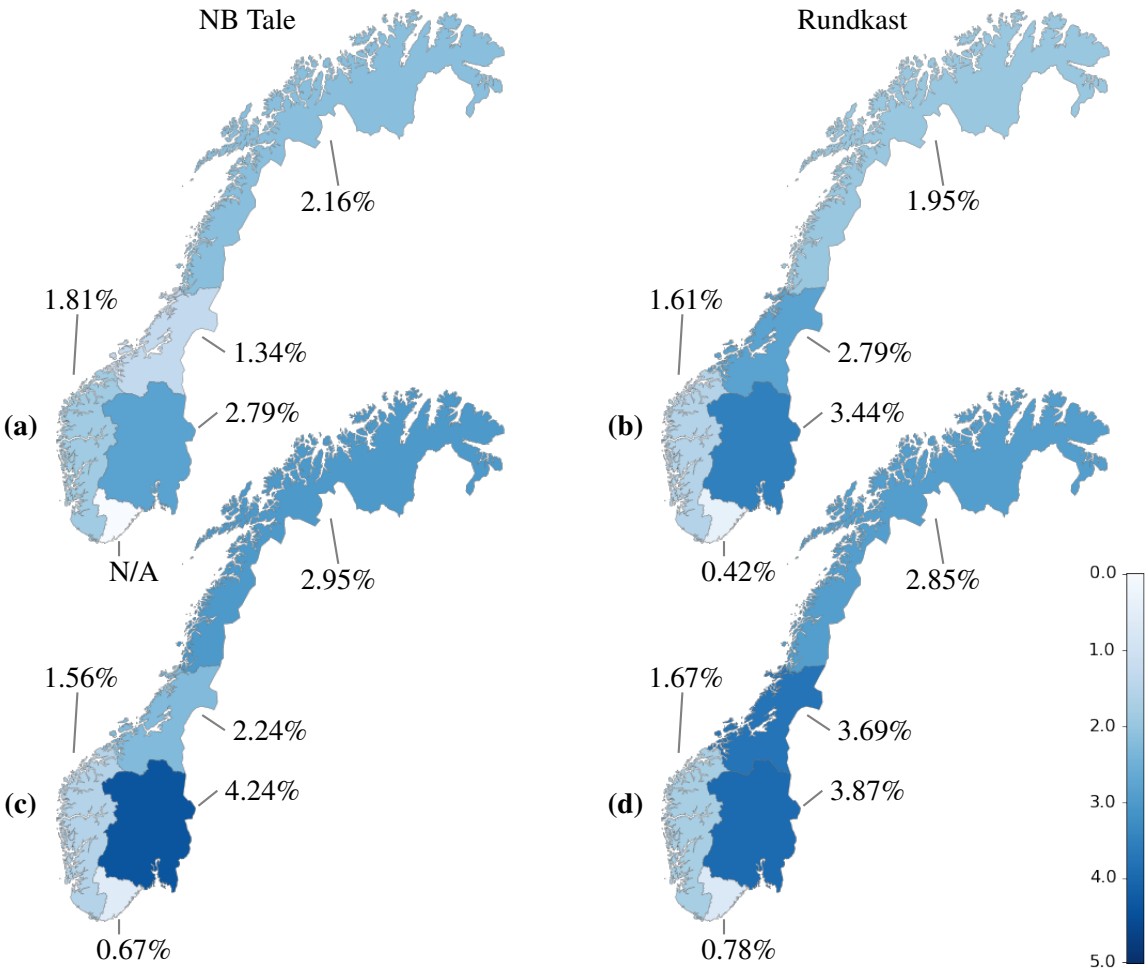

Figure 1: Instances of "r" becoming "l". The first column (a, c) show results on the NB Tale utterances; second column (b, d) shows results on the Rundkast utterances.The first row (a, b) being results from the 300m model and the second row (c, d) being results from the 1b parameter model.

the model. The former being a brief interruption in the airflow and the latter being a continuous, smooth approximant. However, for speakers in the "East" and "Mid" parts of the country, where both the tapped [ɾ] and flapped [ɽ] dialect features are present, we would anticipate a greater degree of confusion. Both tapped [ɾ] and flapped [ɽ] are seen as brief closures with acoustic differentiation relegated to the F3 and F4 trajectories (Kvale and Foldvik, 1995).

Therefore to evaluate how much of an impact these potentially similar realizations have on the model, we used the aligned Bokmål texts (as described in Section 3.2) and calculated how frequently "r" was transcribed instead of "l" and vice versa. When analyzing instances of "r" transforming into "l", we only considered instances where the "r" did not precede another alveolar consonant ("t", "d", "n", "l", "s"). This is due to the fact that

"r", when followed by an alveolar consonant, can be interpreted as a digraph. In dialect regions with the alveolar [ɾ], speakers will realize the second alveolar consonant as a retroflex instead of pronouncing two distinct sounds. That is, "rt" would be realized as [ʈ]). To ensure these realizations did not cloud our analysis, we excluded all "r"s followed by an alveolar consonant.

The maps in Figures 1 and 2 show the percentage of error. That is, for those instances where an "r" was not transcribed correctly, the maps show what percentage of those errors were because an "l" was transcribed instead (Figure 1). And vice versa for the "l" to "r" transformation (Figure 2). This error calculation and plotting was done for each of the cardinal dialect region. Darker colors represent higher errors. In both figures the first column (a, c) show results on the NB Tale utterances; second column (b, d) shows results on the

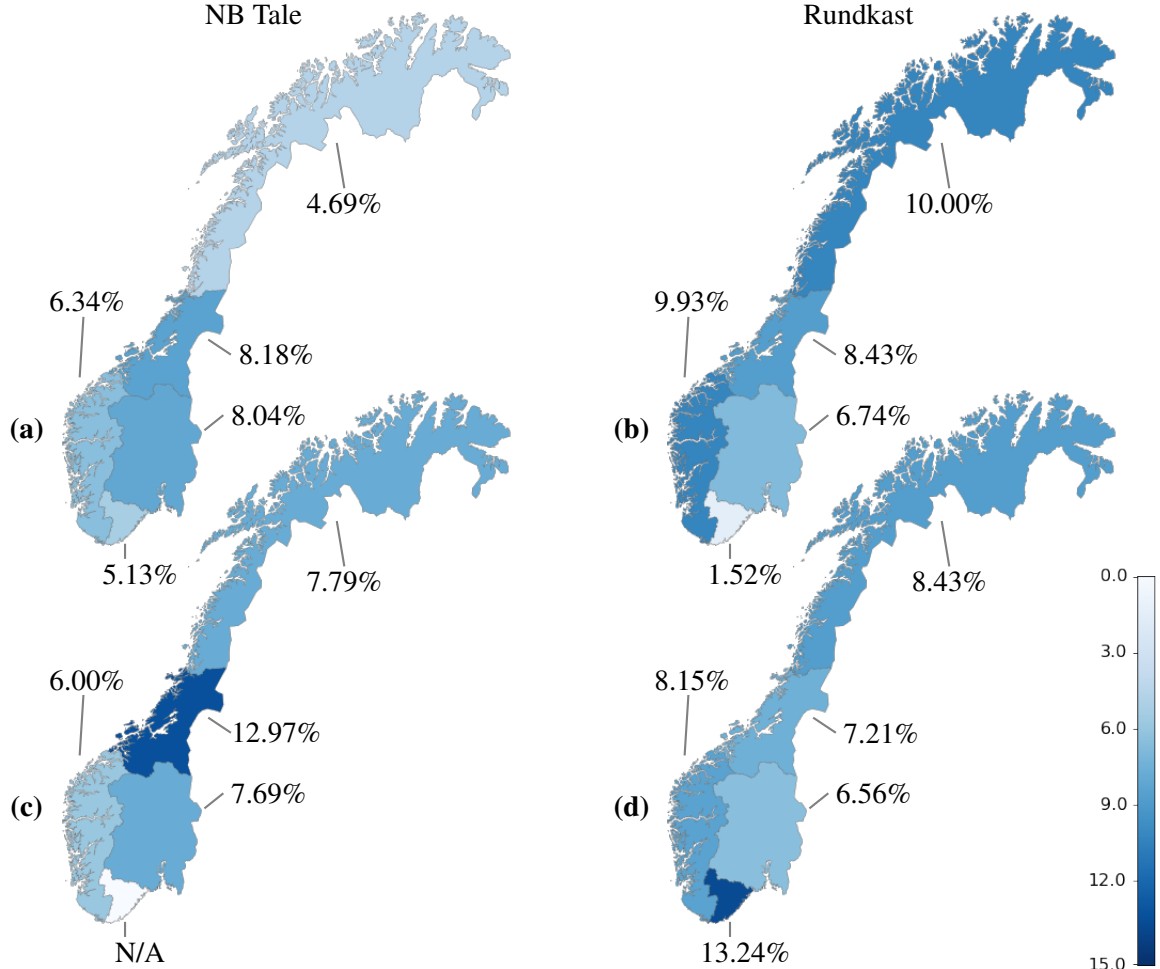

Figure 2: Instances of "l" becoming "r". The first column (a, c) show results on the NB Tale utterances; second column (b, d) shows results on the Rundkast utterances.The first row (a, b) being results from the 300m model and the second row (c, d) being results from the 1b parameter model.

Rundkast utterances. The first row (a, b) being results from the 300m model and the second row (c, d) being results from the 1b parameter model.

For all Figures, except 2(b) and 2(d), the regions with the most confusability between "r" and "l" are the "East", "Mid", and "North". Indeed, for all Figures except 2(d) the "South" has the lowest incidences of "r" and "l" confusion. By and large we also see much clearer, more consistent trends with the NB Tale data. This could be because the utterances in the NB Tale corpora were selected for phonological coverage and thus there were more environments for "r" and "l" confusion.

### 4.3   Voiceless stop lenition

In addition to /r/ and /l/ confusability, we also investigated the distribution of voiceless stop consonants. In the "South" region, voiceless stops tend to lenite to their voiced counterparts in post-

vocalic environments (Skjekkeland, 1997). Thus, we would expect [p], [t], and [k] to lenite to [b], [d], and [g] when preceded by a vowel. To understand if this change is captured by the wav2vec model, we found instances where a voiceless stop was changed and then ensured that the change was to its voiced counterpart. If a voiceless to voice change occurred, we then ensured that both the voice and voiceless stops were preceded by a vowel. We counted occurrences of this postvocalic voicing change across all three stops of interest. Results can be see in Figure 3 for the NB Tale data and Figure 4 for Rundkast. The first column (a) shows results from the 300m parameter model, second column (b) shows results from 1b parameter model. Darker colors represent higher errors.

For both the NB Tale and Rundkast corpora we,

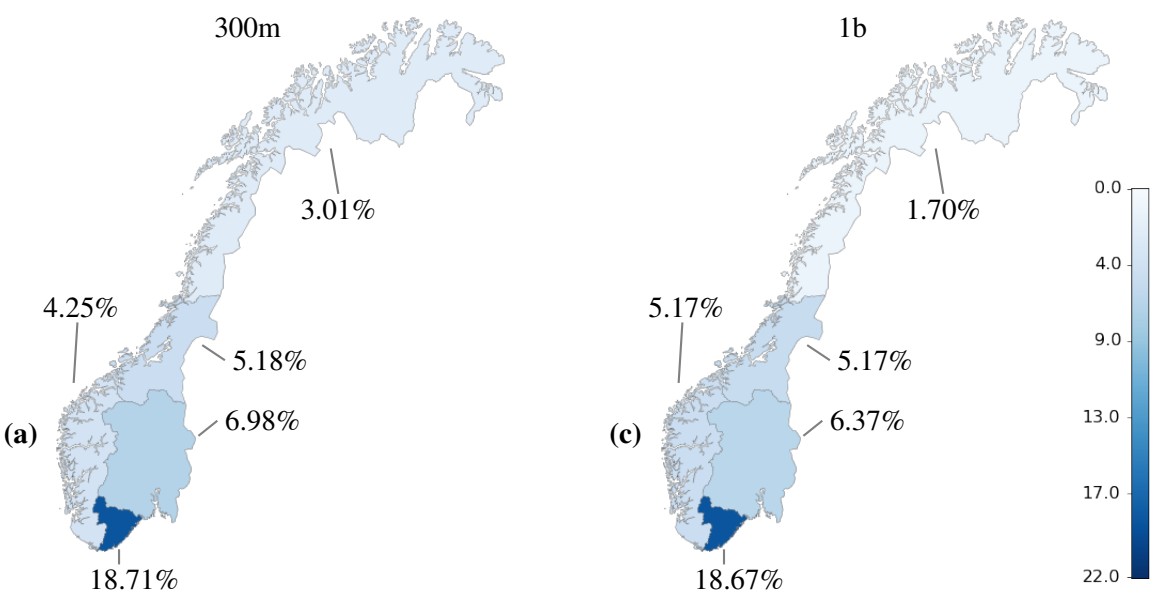

Figure 3: Percentage of postvocalic voicing error; that is, instances of ("p", "t", "k") realized as ("b", "d", "g") as a percentage of total ("p", "t", "k") errors on the NB Tale dataset. First column (a) shows results from the 300m parameter model, second column (b) from the 1b parameter model

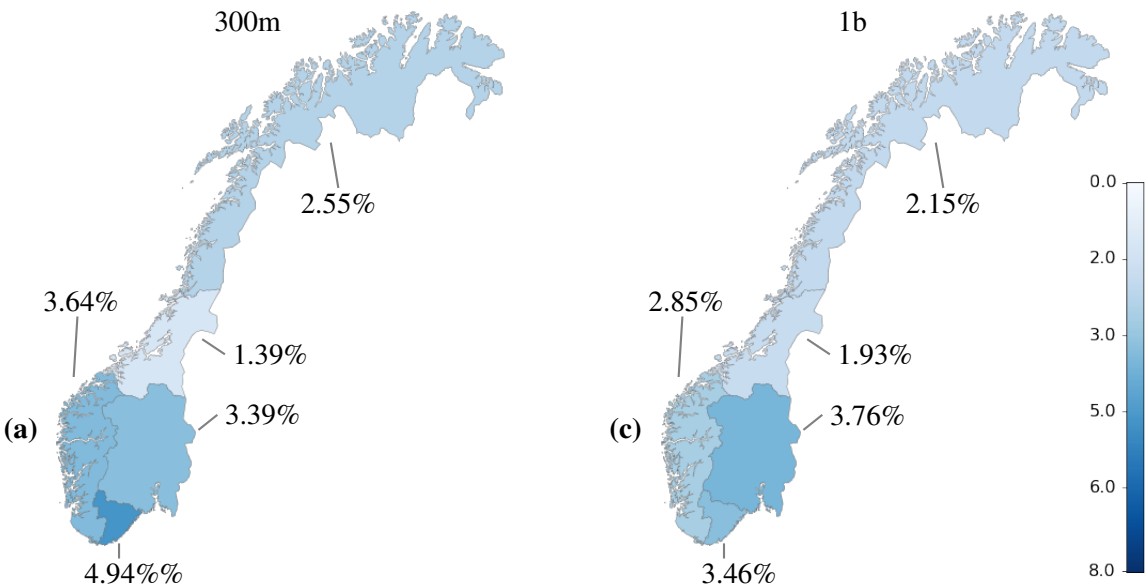

Figure 4: Percentage of postvocalic voicing error; that is, instances of ("p", "t", "k") realized as ("b", "d", "g") as a percentage of total ("p", "t", "k") errors on the Rundkast dataset. First column (a) shows results from the 300m parameter model, second column (b) from the 1b parameter model

can see that the "South" region has the highest instances of voicing. Though once again, we see stronger trends in the NB Tale data then in Rundkast.

## 4.4 Personal pronoun *jeg*

As mentioned when discussing the Norwegian language in Section 2, there are many ways for Norwegain speakers to say the first person pronoun *jeg*. This was briefly investigated as well. Confusion pairs for *jeg* were aggregated and trends sought. Regardless, no trends in the words substituted for *jeg* in the transcripts could be found. This lack of results could indicate that a word like *jeg* occurs so frequently in all dialects that there is an abundance of training examples for the model to generalize from. Or, perhaps, the 5-gram language model used, in addition to the wav2vec component, had enough influence to ensure that only *jeg* was produced.

## 5 Discussion

Due to the fact that we have been able to largely see acoustic dialectal features surfacing through our analysis, we find that this method of carefully aligning text and aggregating results has promise. Furthermore, we infer that the models have learned enough about Norwegian to understand standard spellings and apply these generalizations to broader contexts. Additionally, the phonetic information in the dialects is strong enough to cause the models to utilize this general spelling knowledge and create more acoustically aligned outputs. However, going so far as to say that the models have internalized some knowledge about the dialects themselves (e.g., phonetic features) is perhaps more than can be reasonably asserted from this analysis.

Through this paper we have explored a couple of known dialectally-motivated phonological realizations. There still, however, exist more that could be explored. As mentioned in Section 4.2, there exists a pattern of retroflexting of alveolar consonants for certain Norwegian dialects. This analysis could certainly be extended to those environments. However, there are also phonological changes that are hard, or potentially impossible to see in spelling changes. For example, alveolars are palatalized (most strongly) in the "Mid" region as well as in certain phonological environments in the "North" and the northern parts of the "West"

and "East" regions. This palatalization would be hard to see in spellings since there is no standard way in Norwegian orthography of representing a palatalized sound. Additional Norwegian phonological features that have no written representation (such as toneme) would also be invisible to the analysis performed in this paper.

As the NPSC is derived from parliamentary speeches, the distribution of parliament speakers emulates the population distribution of the country. Thus our models, all of which were trained on NPSC, have the same speaker representation. That is, the "East" region would be the most represented in the training data. Given this, and the results in Table 5, it would seem that the models have best learnt the features which they saw the most, as machine learning models are wont to do. Therefore, if models are to be robust against dialects, it seems necessary to increase the training data for the other regions. Additionally, it might be possible to assign greater weight to these dialectal character changes during training to encourage the models to learn a better representation.

## 6 Conclusion

Through this paper, we demonstrate how an analysis of character errors in transcriptions generated by an end-to-end ASR system can contain dialectal trends mirroring those known through linguistic descriptions. We showed increased confusability between "r" and "l" in regions where those phonemes are realized similarly. We also showed increased incidences of voiceless stop lenition in a region known for that phenomena. These errors indicate that the end-to-end system has successfully learnt to spell in Norwegian, going so far so as to slightly spell in dialect.

## 7 Acknowledgements

This work has been done as part of the SCRIBE project as funded by the Norwegian Research Council, project number: 322964.

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
