# OpenReview forum: "A character-based analysis of impacts of dialects on end-to-end Norwegian ASR"
_NoDaLiDa/2023/Conference — NoDaLiDa 2023_

### Official Review · Reviewer_9GFQ · 2023-02-25
**The authors of this paper proposes a method of correcting ASR based on characters for Norwegian. Their approach is successfull.**

**Rating:** 8
**Confidence:** 3

**Review:**

Please explain your method with an example / or a flow chart
"However, in our solution instead of a fixed cost for substitutions, we allow it to be dynamically computed as
the character error rate (CER)"

Can you compare your approach to other approaches, is it better? Is there any take home message?

Fix reference capitalisation

Per Erik Solberg, Pablo Ortiz, Phoebe Parsons, 942 Torbjørn Svendsen, and Giampiero Salvi. Improving generalization of norwegian asr with limited linguistic resources. Submitted for publication.
=>
Per Erik Solberg, Pablo Ortiz, Phoebe Parsons, 942 Torbjørn Svendsen, and Giampiero Salvi. Improving generalization of Norwegian asr with limited linguistic resources. Submitted for publication.


**Paper Type:**

Long paper

---

### Official Review · Reviewer_98P1 · 2023-03-01
**This paper investigates how certain dialectal features impact the performance of ASR systems for Norwegian**

**Rating:** 7
**Confidence:** 4

**Review:**

This paper aims to investigate how certain dialectal features impact the performance of ASR systems. The authors compare three pre-trained Norwegian wav2vec2 models on three datasets, five dialect areas and three dialectal features. This rather large set of experimental variables makes the paper a bit difficult to follow, but overall, the goals, methods and results are well explained. While the character alignment method is not particularly original, it is well suited for the task. The results are limited to two variables that can be expressed by single graphemes, r/l confusion and voiceless stop lenition, but they aptly show the potential of the approach.

Questions:
- L226: does the "5-gram language model" refer to characters or words?
- L270: do you also use the non-native speakers for your experiments? Is the childhood municipality available for all of them?
- §3.2: If I understand correctly, your approach consists of two steps, word alignment (with Levenshtein distance) and character alignment within aligned words (CER). Wouldn't it work equally well to compute CER directly on the entire utterance/sentence?
- §3.2: Can you share some more details of your phonetic feature vector design? For example, the `epitran.vector` library provides conversion routines for a wide range of languages (although not for Norwegian). Also, the general idea of comparing characters based on their pronunciation has been explored thoroughly by the Groningen dialectometry team in the early 2000s (Nerbonne, Heeringa, Wieling).

Typos and minor remarks:
- L036: system*s* + add full stop
- L092: *the* language model
- L096: such as Salvi did in (Salvi...) > (Salvi...)
- L121: capture*d*
- L154: model.In (add space)
- L163: Thus illustrating > Thus we illustrate
- L175: both > all of
- L198: "quite large" - it is probably hard to determine a precise number anyway
- §3.1: Experimental setup *and data*
- L219: Some words missing in the sentence
- L235: an excellent candidate > excellent candidates
- L255: were created ... during corpus creating > were added ... during corpus creation
- L282: capital*i*zations
- L379: add full stop
- Figure 1: Please use the same scale for all graphs - currently the 4.24% of (b) are darker than the 8.04% of (c) and the 10.00% of (g). It might also help to add the dialect region names to one of the maps (although the cardinal directions should be self explanatory).
- Table 3: What happened to the Nynorsk Unknown utterances? WER of 0 would mean everything correct?
- L565: Nynorks > Nynorsk
- L624: proceed > precede
- L841: postvocallic > postvocalic
- Regarding the presentation of the results in general, I wonder if tables would not be easier to grasp than the maps. I find it a bit hard to get the gist of your experiments with so many maps at the same time. You could still show one or two maps as examples.


**Paper Type:**

Long paper

---

### Official Review · Reviewer_xJdL · 2023-03-06
**Can one identify Norwegian dialects from the output of a standard state of the art ASR system? Unclear if one can, and unclear if one should care.**

**Rating:** 4
**Confidence:** 4

**Review:**


This paper investigates the following question: can one identify
Norwegian dialects from the output of a standard state of the art ASR
system?

Dialectal variation and identification is an interesting topic, and
leveraging state of the art models should always be a goal. This paper
is therefore quite relevant for the venue and original enough.
Unfortunately I see three serious flaws with the present paper.

1. To actually answer the scientific question that you pose, you
   should establish a 'ground truth' (say, what is the linguistically
   established rate of Voiceless stop lenition per region) and compute
   a correlation coefficient (or some other metric) with what the
   model computes. In the present state of the paper, the reader must
   already be aware of the regional Norwegian dialects and perform a
   correlation by themselves to answer the question.


2. The methodology is somewhat roundabout. Why not use a speech to
   phoneme model instead of speech to text? Current methodology seem
   weirdly dependent on the idiosyncrasies of the writing systems and
   the exact peculiarities of the models employed (which does suppress
   dialectic differences--- exactly what you observe with the 1st
   person pronoun experiment.)

   The natural reaction of many readers will be to ask if this method
   is promising or not. You should strive to answer this question by
   comparing your method to prior art (some of which you cite) and
   models which output phonemes.

3. The authors do not convey thorough understanding of some of the
   methods that they employ (alignment, see below).


Reading notes:


111: "has been obscured by the network itself." -> "is not transparent, by design."

154: missing space after dot.

195,202,301,351,etc. Non standard opening double quotes (for English “ is the preferred opening double quotes)

216-220: this sentence appears to be malformed

295: "With traditional, Levenshtein-based alignments, this sort of
word similarity is not considered. Any pair of words that do not
exactly match are treated as completely different." There is such a
thing as character-based Levenshtein alignment. The paper should state
why is it not considered and authors prefer to go a complicated route.

308: why is it dynamic? Dynamic usually means dependent on context,
but surely this measure is only a function of the words encountered?

320: "compute the substitution cost between characters as the
Euclidean distance between two feature vectors." What are those
feature vectors? Typically one would use a cosine
similarity.

Table 1: Speaks about non-character aware, but is also done at a
character level. This is self-contradictory. You mean
not aware of phonetics.

347: "treated equally": you mean that all character differences have
the same cost.

632: "we excluded all possibilities." What does "all" mean here?

797: "machine learning models are wont to do." (?)


**Paper Type:**

Long paper

---

### Decision · Program_Chairs · 2023-03-17

Accept